# RNA-Seq Transcriptome Analysis and Evolution of *OsEBS*, a Gene Involved in Enhanced Spikelet Number per Panicle in Rice

**DOI:** 10.3390/ijms241210303

**Published:** 2023-06-18

**Authors:** Fuan Niu, Mingyu Liu, Shiqing Dong, Xianxin Dong, Ying Wang, Can Cheng, Huangwei Chu, Zejun Hu, Fuying Ma, Peiwen Yan, Dengyong Lan, Jianming Zhang, Jihua Zhou, Bin Sun, Anpeng Zhang, Jian Hu, Xinwei Zhang, Shicong He, Jinhao Cui, Xinyu Yuan, Jinshui Yang, Liming Cao, Xiaojin Luo

**Affiliations:** 1State Key Laboratory of Genetic Engineering and MOE Engineering Research Center of Gene Technology, School of Life Sciences, Fudan University, Shanghai 200438, China; niufuan224@126.com (F.N.); 22210700009@m.fudan.edu.cn (J.C.);; 2Key Laboratory of Germplasm Innovation and Genetic Improvement of Grain and Oil Crops (Co-Construction by Ministry and Province), Ministry of Agriculture and Rural Affairs, Crop Breeding and Cultivation Research Institute, Shanghai Academy of Agricultural Sciences, Shanghai 201403, Chinasunbin@saas.sh.cn (B.S.);

**Keywords:** rice, spikelet number per panicle, transcriptome, haplotype, evolution

## Abstract

Spikelet number per panicle (SNP) is one of the most important yield components in rice. Rice *ENHANCING BIOMASS AND SPIKELET NUMBER* (*OsEBS*), a gene involved in improved SNP and yield, has been cloned from an accession of Dongxiang wild rice. However, the mechanism of *OsEBS* increasing rice SNP is poorly understood. In this study, the RNA-Seq technology was used to analyze the transcriptome of wildtype Guichao 2 and *OsEBS* over-expression line B102 at the heading stage, and analysis of the evolution of *OsEBS* was also conducted. A total of 5369 differentially expressed genes (DEGs) were identified between Guichao2 and B102, most of which were down-regulated in B102. Analysis of the expression of endogenous hormone-related genes revealed that 63 auxin-related genes were significantly down-regulated in B102. Gene Ontogeny (GO) enrichment analysis showed that the 63 DEGs were mainly enriched in eight GO terms, including auxin-activated signaling pathway, auxin polar transport, auxin transport, basipetal auxin transport, and amino acid transmembrane transport, most of which were directly or indirectly related to polar auxin transport. Kyoto Encyclopedia of Genes and Genomes (KEGG) metabolic pathway analysis further verified that the down-regulated genes related to polar auxin transport had important effects on increased SNP. Analysis of the evolution of *OsEBS* found that *OsEBS* was involved in the differentiation of *indica* and *japonica*, and the differentiation of *OsEBS* supported the multi-origin model of rice domestication. *Indica* (*XI*) subspecies harbored higher nucleotide diversity than *japonica* (*GJ*) subspecies in the *OsEBS* region, and *XI* experienced strong balancing selection during evolution, while selection in *GJ* was neutral. The degree of genetic differentiation between *GJ* and *Bas* subspecies was the smallest, while it was the highest between *GJ* and *Aus*. Phylogenetic analysis of the Hsp70 family in *O. sativa*, *Brachypodium distachyon*, and *Arabidopsis thaliana* indicated that changes in the sequences of *OsEBS* were accelerated during evolution. Accelerated evolution and domain loss in OsEBS resulted in neofunctionalization. The results obtained from this study provide an important theoretical basis for high-yield rice breeding.

## 1. Introduction

Rice (*Oryza sativa*) is one of the most important food crops worldwide; indeed, more than half of the global population relies on rice as their staple food [1]. Spikelet number per panicle (SNP), one of the three components of yield, is an important agronomic trait in rice. To date, more than 40 genes involved in this trait have been cloned; these genes are distributed among 10 of the 12 rice chromosomes (the 10th and 12th chromosomes being the exceptions). Among these genes, *Gn1a*, a negative regulatory factor, was the first major quantitative trait locus (QTL) cloned that controlled SNP in rice [2]. Rice *DENSE AND ERECT PANICLE 1* (*OsDEP1*) promotes cell division and increases the number of branches, resulting in increased rice SNP [3,4]. Rice *SQUAMOSA PROMOTER BINDING PROTEIN-LIKE 14*/*Ideal Plant Architecture 1* (*OsSPL14/IPA1*) and *OsSPL18* bind to the *OsDEP1* promoter during the reproductive growth stage and positively regulate the expression level of *OsDEP1*, resulting in a larger panicle size and increased SNP in rice [5,6,7,8]. Recent studies revealed that *LARGE2* regulates panicle size and grain number in rice by encoding a Homologous to E6AP C-Terminus (HECT)-domain E3 ubiquitin protein ligase (UPL) OsUPL2 [9]. Many SNP genes, such as *Ghd7* and *PRL5* [10,11], have multiple effects. For example, *Ghd7* not only controls SNP, but also produces a significant impact on plant height and heading period in rice.

The development of a rice spikelet involves a series of complex events. At the reproductive growth stage, the shoot apical meristems of rice expand sequentially and become the inflorescence meristem, branch meristem, spikelet meristem, and flower meristem [12,13]. This process is often influenced by endogenous hormones. Auxin, cytokinin (CTK), gibberellin (GA), abscisic acid (ABA), ethylene (ETH), and brassionolide (BR) are currently recognized as six major categories of plant hormones. Previous research has conclusively demonstrated that auxin is a determinant of plant architecture, and polar auxin transport (PAT) plays a key role in the regulation of many aspects of plant growth and development [14,15]. Rice *PINOID* (*OsPID*) and *PLANT ARCHITECTURE AND YIELD 1* (*OsPAY1*) can change spikelet structure by affecting PAT and have an important impact on SNP in rice [16,17]. There is also a significant association between spikelet development and CTK content in rice. High concentrations of CTK/indole-3-acetic acid (IAA) are beneficial for prolonging the differentiation period and promoting spikelet differentiation during spikelet development [18,19,20]. Over-expression of rice *Auxin-signaling F*-Box 6 (*OsAFB6*), the gene that encodes an auxin signal transduction factor, causes a significant decrease in IAA level and a significant increase in the biological activity level of CTK, leading to a larger spike [21]. In addition, ETH, ABA, and BR also affect rice SNP [22,23,24].

A haplotype is defined as the combination of multiple single nucleotide polymorphisms in linkage disequilibrium on the same chromosome. Compared to single nucleotide polymorphism analysis, haplotype analysis can provide more extensive polymorphisms, which is more conducive for analysis of the origin and function of genes at a population level. Sequences from the completed 3000 Rice Genome project provided more than 32 million single nucleotide polymorphisms and more than 93,000 genome structural variations for genome research [25,26], offering a rich source of data for rice population genetic analysis.

The rice gene *ENHANCING BIOMASS AND SPIKELET NUMBER* (*OsEBS*), involved in SNP and biomass, was previously cloned from Dongxiang wild rice (*O. rufipogon Griff*). Sequence alignment analysis showed that OsEBS contains a region with high similarity to the N-terminal conserved ATPase domain of heat shock protein 70 (Hsp70) but lacks the C-terminal regions of the peptide-binding domain and the C-terminal lid [27]. However, the mechanism responsible for increased SNP associated with *OsEBS* is still unclear. In this study, RNA-Seq transcriptome analysis was used to identify the potential regulatory mechanism of OsEBS for increasing SNP, and analysis of the evolution of *OsEBS* was performed. The results obtained from this study provide a theoretical basis for a high-yield rice breeding strategy.

## 2. Results

### 2.1. RNA Sequencing and Sequence Alignment of Guichao 2 and B102

The cDNA libraries of six young panicle samples from two rice accessions were sequenced. A total of 255,089,536 clean reads were obtained with an average of 42,514,923 reads per sequenced library. The clean Q30 Bases Rate was >94% for all six cDNA libraries (Table 1). The only alignment sequences were mostly aligned to the exon region (Appendix A), and the sequencing data had uniform coverage across the entire genome (Appendix A). The probability of clean reads mapping to the reference genome ranged from 85.70% to 87.12%, with an average of 86.46% (Appendix A). These results indicated that the transcriptome sequencing quality was relatively high and suitable for subsequent data analysis.

### 2.2. Analysis of Differentially Expressed Genes (DEGs) in the Panicle of Guichao 2 and B102

Significant DEGs were defined as having a q value < 0.05 and an absolute log2-fold change (FC) ≥ 1. A total of 5369 DEGs were detected between B102 and Guichao2 during spikelet development, and the DEGs were mainly down-regulated in B102 (Figure 1A,B, Appendix A). Plant hormones play extremely important roles in the regulation of plant growth and development. Numerous studies have shown that plant hormones can affect rice SNP by regulating molecular factors and their interactions [28,29,30,31,32]. We analyzed the expression of endogenous hormone genes in rice and found that many auxin-related genes were differentially expressed between Guichao 2 and B102; 14 genes were significantly up-regulated and 63 genes were significantly down-regulated in B102. Further analysis revealed that the magnitude of FC for the 14 up-regulated genes was small. In contrast, the magnitude of FC for the 63 down-regulated genes was relatively large; in particular, the FC of *OsZS_01G0558800* was over five (Appendix A). The above results suggested that the down-regulated expression of auxin-related genes might play an important role in the formation of B102 spikelets.

### 2.3. Function Enrichment Analysis

To analyze the functional distribution characteristics of DEGs, Gene Ontology (GO) enrichment analysis was performed for biological processes (BPs), cellular components (CC), and molecular functions (MF). The results showed that many DEGs participated in BP such as response to stimulus, response to stress, biosynthetic process, and transport during spikelet development. For CC, DEGs were mainly located in the cell part, the membrane part, membrane and plasma membrane. For MF, DEGs were mainly concentrated in the GO terms such as oxidoreductase activity, transporter activity, tetrapyrrole binding, and transmembrane transporter activity. Further analysis showed that auxin-related DEGs were mainly enriched in eight BP terms, including auxin-activated signaling pathway, auxin polar transport, auxin transport, basipetal auxin transport, and amino acid transmembrane transport, most of which are directly or indirectly related to PAT (Table 2). The above results indicated that DEGs may play an important role in spikelet formation by influencing PAT transport during spikelet development.

The Kyoto Encyclopedia of Genes and Genomes (KEGG) database was used to analyze the metabolic pathways enriched by DEGs. DEGs were mainly enriched in 124 KEGG metabolic pathways, among which 11 pathways were significantly enriched, with most DEGs down-regulated in B102 (Figure 1C, Appendix A). Analysis of the distribution of significantly down-regulated DEGs (Appendix A) revealed that *OsZS_03G0171700* was significantly enriched in the amino sugar and nucleotide sugar metabolism pathway, and *OsZS_02G0351400* in the flavonoid biosynthesis pathway. Notably, *OsZS_01G0558800*, *OsZS_08G0655500*, and *OsZS_05G0046000* were all clearly enriched in the ABC transporters metabolic pathway (q = 0.3994) (Appendix A).

### 2.4. OsEBS Participated in the Differentiation of Indica and Japonica Subspecies

Previous studies found that the distribution of two *OsEBS* alleles, full-length as Guichao2 and truncated as B102, differed between *indica* and *japonica* rice subspecies; the possibility that OsEBS differentiation occurred before common wild rice evolved into cultivated varieties has been suggested [27]. However, fewer rice accessions were used in the previous studies (Appendix A). To further determine whether OsEBS was involved in the differentiation of *indica* and *japonica* subspecies, 173 rice accessions were re-examined (Appendix A). Combined with previous research accessions, it was found that the truncated *OsEBS* allele was present in 85.8% of indica varieties, and the full-length allele was present in 93.0% of *japonica* varieties. The full length and truncated alleles were evenly distributed in wild rice. The above results indicated that genomic sequence variation of *OsEBS* was related to the differentiation of the *indica*–*japonica* subspecies, and *OsEBS* differentiation occurred before wild rice evolved into cultivated varieties.

### 2.5. Population Genetic Analysis

To further analyze and validate the evolutionary relationship of *OsEBS* among rice subpopulations, 11 major gene-coding sequence (CDS) haplotypes (gcHaps) (exist in ≥30 rice accessions) [25] were identified based on 12 SNPs distributed among the CDS region of OsEBS in 2751 accessions from the 3K-RG database. *Xian/Indica* (*XI*) accessions predominantly contained Hap2, Hap3, Hap4, Hap5, and Hap7, whereas *Geng/Japonica* (*GJ*) accessions predominantly carried only Hap1 (Figure 2A,B, Appendix A). *XI* and *GJ* accessions were also clearly divided into different haplotypes based on the 1-kb promoter haplotype (gpHaps) analysis (Figure 2C, Appendix A), indicating strong differentiation of *OsEBS* haplotypes between *XI* and *GJ*. However, in contrast to gcHaps, *GJ* accessions were generally divided into more than one gpHap, which may indicate that marker polymorphisms of *OsEBS* were more abundant in the promoter region than in the CDS region in *GJ* accessions.

Analysis of the nucleotide diversity (π) and Tajima’s D statistics of the XI and GJ subpopulations in the OsEBS flanking region revealed the highest nucleotide diversity in the XI subpopulation and the lowest in the Aus subpopulation with π values of XI > Bas > GJ > Aus for the four subpopulations. Moreover, the nucleotide diversity of XI was significantly higher than that of GJ (Figure 3A). In addition, XI showed a positive peak in Tajima’s D in the 2-Mb genome region flanking OsEBS (Figure 3B), indicating that OsEBS was under strong balancing selection in XI, decreasing the frequency of rare alleles in XI. Tajima’s D of GJ was close to 0, indicating that selection on OsEBS in GJ may have been neutral. In addition, results suggested rapid population expansion or directional selection for Aus. Further analysis was conducted on the nucleotide diversity (π) and Tajima’s D statistics of the four XI subgroups in the 2-Mb OsEBS flanking region. The XI-1B subgroup had the most abundant nucleotide polymorphisms in this region (Figure 3C), and the strong balancing selection in XI mainly came from the XI-1B subgroup (Figure 3D).

Tajima’s D analysis based on single nucleotide polymorphisms was conducted using the *OsEBS* gene and the 1-kb upstream promoter. Results confirmed that *XI* was mainly subjected to stronger balancing selection in the CDS region of the *OsEBS* gene rather than the promoter region (Figure 3E,F). Overall, *XI* subgroups had higher nucleotide diversity than *GJ* subgroups in the *OsEBS* region, and *XI* experienced strong balancing selection during the evolution of *OsEBS*, while selection on *OsEBS* in *GJ* was neutral.

To analyze the degree of differentiation between different subspecies, Fst analysis based on single nucleotide polymorphisms of *OsEBS* was performed (Appendix A). The results showed that *GJ* and *Bas* had the smallest degree of differentiation, followed by *XI* and *Aus*, which indicated the close genetic relationship of *GJ-Bas* and *XI-Aus*. The degree of differentiation between *GJ* and *Aus* was the highest, indicating the most distant genetic relationship. Consistent results were also obtained from the heat map in Appendix A.

### 2.6. Accelerated Evolution and Domain Loss of OsEBS Resulted in Neofunctionalization

To analyze the phylogeny of OsEBS, we collected amino acid sequences of heat shock protein 70 (Hsp70) from *O. sativa*, *Brachypodium distachyon*, and *Arabidopsis thaliana*. We obtained a total of 42 homologous genes, and constructed a phylogenetic tree using the Maximum Likelihood (ML) method with MEGA5 software (V11.0.10). There were seven major phylogenetic clades corresponding to seven subgroups (H1–H7) (Figure 4A). According to the characteristics of these sequences, H5, H6, and H7 were significantly different from other subgroups (Appendix A). Hence, we constructed a phylogenetic tree using the H5, H6, and H7 clades as the outgroup and found H2 to be the sister group of H1. The bootstrap support value for the node joining H1 and H2 was high (71). These results suggested that H2 is more closely related to H1 than to any other subgroup. The same tree topology was obtained using Jones–Taylor–Thornton (JTT) and Whelan and Goldman (WAG) substitution models with seven alignment methods, with the exceptions of the WAG model from MUSCLE and the T-coffee alignment (Appendix A). This result was further supported by specific shared amino acid sites and the similarity scores of pair alignments between H2 and other subgroups (Appendix A, Appendix A).

We predicted the domain distribution of these proteins using InterProScan “www.ebi.ac.uk/Tools/pfa/iprscan/”(accessed on 10 June 2012). Hsp70 proteins usually have four domains: the signal peptide, the ATPase domain, the peptide-binding domain, and the C-terminal lid. The C-terminal lid domain differed significantly among the seven subgroups. The sequence of the C-terminal lid indicated that group H1 was located in the endoplasmic reticulum, H3 in chloroplasts, and H4 in mitochondria. However, the C-terminal lid in groups H6 and H7 remains unidentified. Proteins in the same subgroup shared highly conserved domain components and architecture (Figure 4B). However, there were some distinct differences among subgroups, indicating loss or gain of domains among the subgroups. The domain architecture analysis indicated that the ancestral protein of Hsp70 contained at least an ATPase domain, a peptide-binding domain, and a C-terminal lid. During the evolution of Hsp70 proteins, various domain loss events occurred, mainly the C-terminus and N-terminus domains. The H2 subgroup containing OsEBS is a typical example of an Hsp70 that lost the peptide-binding domain and the C-terminal lid.

To determine whether the H2 subgroup is under different evolutionary constraints than other subgroups, the ω values of the H2 branch and the other branches were calculated using Phylogenetic Analysis by Maximum Likelihood (PAML). Under the multiple frequencies (mF) model, the ω value of the H2 branch was 2.7397, which is much higher than the value from the one ratio (M0) model (0.0935) (Appendix A). This result suggests that the H2 subgroup evolved rapidly in comparison with other subgroups. From the results described above, we concluded that OsEBS is an Hsp70 variant that has many new features that differ from the conserved features in Hsp70s. *OsEBS* differed in expression patterns from those of typical *Hsp70* genes. Decreased expression of two other rice genes (*Os06g10990* and *Os12g05760*) in the H2 subgroups under heat shock treatment was also observed to be similar to *OsEBS* (Appendix A). In summary, OsEBS underwent a rapid change in its amino acid sequence and domain loss, and these changes ultimately led to a novel function.

## 3. Discussion

SNP is an important trait affecting rice yield. In recent decades, breeding practices have shown that improvement of SNP is the most effective way to increase rice yields [33]. In this study, RNA-seq analysis of Guichao 2 and an *OsEBS* over-expression line B102 was conducted to explore the mechanism of *OsEBS* regulation of rice SNP at the heading stage. The results showed that DEGs in the overexpressed line B102 were mostly down-regulated. Plant endogenous hormones are closely associated with the formation of rice SNP. We further compared the expression of endogenous hormone genes between Guichao 2 and B102, and found that many auxin-related genes were significantly down-regulated in B102. GO enrichment analysis showed that DEGs related to auxin regulation were mainly enriched in biological processes related to PAT. KEGG metabolic pathway analysis found that some auxin-related DEGs were enriched in flavonoid biosynthesis and ABC transporters pathways, which are closely related to auxin transport. Flavonoids are secondary metabolites of the phenylpropanoid pathway, which could regulate PAT by directly affecting the distribution, activity, and content of auxin carriers or regulate the transport and distribution of auxin by indirectly affecting the antagonism of protein kinase/phosphatase and cell membrane fluidity among other mechanisms [34,35,36]. The ABC family of proteins are transmembrane transport proteins that can participate in many life processes including plant hormone transport [37]. Among these proteins, members of the ABC subfamily in class B (ABCB) are key components of auxin efflux [38]. Our results demonstrate that DEGs involved in PAT play an important role in the formation of B102 SNP.

Auxin is mainly synthesized in the shoot tip and transported to the base of the stem through the vascular system and finally to the root tip through PAT, which is coordinated by AUXIN1/LIKE-AUX 1 (AUX/LAX), PIN-FORMED (PIN), and ABCB family proteins [39]. PAT is important in the development of young rice panicles [17,40]. Further, the ratio of CTK/IAA has an important effect on organ differentiation. A high ratio of CTK/IAA during rice spikelet differentiation was conducive to promoting panicle differentiation and inhibiting spikelet degeneration [18,19]. For example, over-expression of auxin signal transduction factor OsAFB6 caused a significant decrease in the IAA level and increase in the CTK level, leading to larger spike shapes [21]. Therefore, we speculate that over-expression of the OsEBS gene in B102 caused the down-regulation of genes involved in PAT in the panicle, which led to increased CTK/IAA, promoted the differentiation of secondary branches, and thus increased SNP. The molecular mechanism of SNP regulation by OsEBS will be more clearly revealed by further investigation of metabolic pathways and their underlying genes. The study of the molecular mechanism of OsEBS control of SNP has further enriched our understanding of the mechanism of auxin transport in rice.

At present, the origin and evolution of cultivated rice are still a focus of scholars worldwide, and there is still great controversy over whether cultivated rice has a primary or secondary origin [41]. Currently, there are three main views on the domestication of cultivated rice. The first view suggests that the *indica* subspecies was directly domesticated from ordinary wild rice, while the *japonica* subspecies was domesticated from *indica*. The second view suggests that rice domestication has a multi-origin model, and *indica* and *japonica* subspecies were domesticated from different wild rice lineages. The third view suggests that rice domestication has a single origin model, and the two subspecies, *indica* and *japonica*, originated from the same ancestor of wild rice. We found that genomic sequence variation of OsEBS was related to the differentiation of the indica and japonica, and the differentiation of *OsEBS* occurred before wild rice evolved into cultivated varieties. Therefore, *OsEBS* differentiation supports the multi-origin model for rice domestication. The *OsEBS* full-length allele from Dongxiang wild rice has been demonstrated to improve SNP and rice yield, but it mainly exists in *japonica* varieties. Due to its participation in the differentiation of *indica* and *japonica*, it is rarely used in *indica* rice at present. Introgression of the *OsEBS* full-length allele into high-yielding *indica* varieties through molecular marker-assisted selection may be an important strategy to further improve the yield of *indica* rice. The results of this study provide a theoretical basis for high-yield rice breeding.

There is intense interest surrounding the evolution of the Hsp70 protein family. Phylogenetic analyses have shown that the Hsp70 family is broadly and highly conserved across prokaryotes and eukaryotes [42,43]. For example, residues are 50% identical between the DnaK from *Escherichia coli* and eukaryotic Hsp70s, and there is 50% to 98% similarity among eukaryotic Hsp70s [44]. There have been a few reports of domain loss facilitating neofunctionalization during the evolution of gene families, such as *OsEBS* [45,46,47]. Interestingly, many studies on domain loss events showed that versatile domains occur more frequently at the C terminus or N terminus [48]. Phylogenetic analysis suggested that the sequence of OsEBS changed rapidly. This finding is consistent with our results, which suggested that only N-signal peptide or C-terminal domain loss or gain events occurred among the different subgroups. Overall, our results mentioned above imply that domain loss is an effective pathway to neofunctionalization. The *OsEBS* gene showed further evidence that domain loss is an important pathway for generating new genes among gene families.

## 4. Materials and Methods

### 4.1. Plant Materials and Growing Conditions

The experimental materials included Guichao 2 (truncated genotype of *OsEBS*) and *OsEBS* over-expression line B102 (full-length genotype of *OsEBS*) with Guichao 2 as the background. Compared with Guichao 2, the transgenic line B102 showed increased plant height (14.35%), bigger biomass (36.08%), and greater number of spikelets per panicle (19.68%) and increased grain yield per plant [27]. In this study, 173 rice accessions, including 114 *japonica* varieties, 55 *indica* varieties, and 4 wild rice accessions, were utilized. In 2021, Guichao 2 and B102 were planted in the experimental field of the Life Science College of Fudan University (Shanghai, China, 121.5° E, 31.3° N, altitude: 4 m, annual average sunshine: 1387 h, annual average temperature: 15.8 °C, annual average rainfall: 1078.1 mm, and annual average evaporation: 1346.3 mm). Concurrently, the 173 rice accessions were planted at the Zhuanghang Comprehensive Experimental Station of Shanghai Academy of Agricultural Sciences (121.4° E, 30.9° N) and were sown on 20 May and transplanted at the stage of 4 leaves unfolded (BBCH scale).

### 4.2. DNA Extraction and Genotype Identification

During the tillering stage, 100 mg fresh rice leaves was collected and thoroughly ground with liquid nitrogen. The ground powder was used for genomic DNA extraction using the CTAB method [49]. A NanoDrop 2000 (Thermo Fisher, Wilmington, DE, USA) spectrometer was used to determine DNA concentration. The primer sequences used for *OsEBS* genotype identification were as follows: *OsEBS-F*: 5′-tggggatttctaggaccgtg-3′, *OsEBS-R*: 5′-gtcccggatgacgaacttg-3′. The primers were synthesized by Shanghai Jieli Biotechnology Co., Ltd., Shanghai, China. PCR amplification reaction mixtures (10 μL) consisted of 1 μL template DNA, 1 μL primer, 5 μL Taq mix and 3 μL ddH_2_O. The PCR products were genotyped using a 1% agarose gel electrophoresis.

### 4.3. Total RNA Extraction and cDNA Library Construction

Samples of Guichao 2 and B102 were strictly taken in the middle area of each group at the stage of beginning of flowering (BBCH scale), with three young panicle samples numbered G21, G22, and G23 (three biological replicates of Guichao 2) and B1021, B1022, and B1023 (three biological replicates of B102). The entire panical was ground for RNA extraction. Total RNA was extracted using an RNAprep Pure Plant Kit (Tiangen, Shanghai, China) following the manufacturer’s instructions. RNA purity, concentration, and integrity were determined using the NanoDrop 2000 spectrometer (Thermo Fisher, Wilmington, DE, USA), Qubit3.0 (Life Technologies, Carlsbad, CA, USA), and Agilent 2100 (Agilent Technologies, Carlsbad, CA, USA), respectively. A total of 2 μg RNA per sample was used for the RNA sample preparations. Sequencing libraries were generated using a NEBNext*^®^* Ultra™ RNA Library Prep Kit for Illumina*^®^* (#E7530L, NEB, Ipswich, MA, USA) following the manufacturer’s recommendations, and index codes were added to attribute sequences to each sample.

### 4.4. RNA Sequencing (RNA-seq), Data Filtering, and Sequence Alignment

The effective concentration of the library was quantified. The high-quality library was sequenced using an Illumina Novaseq 6000 platform with the PE150 sequencing strategy, resulting in 150 bp paired-end reads. After the raw data were acquired, FastQC (V0.11.9) was used for data quality detection and evaluation to obtain clean data. The Fastp (V020.0) software was used to remove joint contaminated and low-quality reads from the raw sequencing data. Reference genome annotation files were downloaded from the ENSEMBL website (http://plants.ensembl.org/index.html, last accessed on 28 April 2023). Bowtie2 v2.2.3 was used for building the genome index, and the clean data were then aligned to the reference genome (ZS97RS3, RIGW3.0) using HISAT2 (v2.1.0) [50]. Data with a match rate of more than 80% were considered well matched and available for further analysis.

### 4.5. Gene Differential Expression Analysis

Read counts for each gene in each sample were counted by HTSeq v0.6.0, and the Fragments Per Kilobase Million Mapped Reads (FPKM) method was then used to calculate the expression level of genes in each sample [51]. DESeq2 was designed for differential gene expression analysis between B102 and Guichao2 with biological replicates. Genes with a q value ≤ 0.05 and FC ≥ 1 were identified as differentially expressed genes (DEGs).

### 4.6. Function Enrichment Analysis

Gene Ontology (GO, http://geneontology.org/, last accessed on 28 April 2023) enrichment of DEGs was conducted using the hypergeometric test, in which the *p*-value was calculated and adjusted to the q value. The data background was genes in the whole genome. Kyoto Encyclopedia of Genes and Genomes (KEEG, http://www.kegg.jp/, last accessed on 28 April 2023) enrichment of DEGs was also conducted using the hypergeometric test. GO and KEGG terms with a q value < 0.05 were considered significantly enriched.

### 4.7. Population Genetic Analysis

The gene CDS haplotypes (gcHaps) and gene-promoter haplotypes (gpHaps) of *OsEBS* were downloaded from RFGB (https://www.rmbreeding.cn, last accessed on 28 April 2023). The R package ggplot2 was used to visualize the multiple comparisons and the distribution of gcHaps among the 5 major populations previously reported by Wang et al. [52] via boxplots and stacked bar plots. The SNP dataset used for calculating nucleotide diversity (π) and Tajima’s D were downloaded from the Rice SNP-Seek Database (https://snp-seek.irri.org/_download.zul, last accessed on 28 April 2023). Nucleotide diversity (π), Tajima’s D, and Fst for the target region were calculated by VCFtools (v.0.1.15).

### 4.8. Sequence and Phylogenetic Analyses

Sequences for predicted proteins in *A. thaliana* and rice were retrieved from the TAIR10 Genome Release (www.arabidopsis.org, last accessed on 28 April 2023) and Rice Genome Annotation Project (V6.1) (rice.plantbiology.msu.edu, last accessed on 28 April 2023), respectively. Sequences for proteins of other plants were obtained from Phytozome v7.0 (www.phytozome.net, last accessed on 28 April 2023). The hidden Markov model-based HMMER program (2.3.2) [53] was used to identify all proteins containing an Hsp70 domain. The Hsp70 domain (PF00012 in the Pfam database) was used to perform local searches in the downloaded proteome datasets. Phylogenetic relationships were estimated using the Maximum Likelihood (ML) method. Because the use of different sequence alignments and substitution models may influence the results of phylogenetic analyses, we conducted a comprehensive sensitivity analysis. First, we aligned sequences using 7 different methods, including Dialign, Kalign2, Mafft, Muscle, T-Coffee, ClustalW2, and Clustal Omega by EBI tools (Multiple Sequence Alignment: www.ebi.ac.uk/Tools/msa/, last accessed on 28 April 2023). Second, for each resulting alignment, we constructed ML trees using JTT and WAG substitution models with 100 bootstraps by MEGA5 [54]. To test for variation of the ω ratio among different branches in the resulting trees, a branch-specific model analysis was conducted in Codeml from the PAML package (v4.4) [55]. Structural motif annotation was performed using the InterProScan (www.ebi.ac.uk/Tools/pfa/iprscan/, last accessed on 28 April 2023) databases.

## Figures and Tables

**Figure 1 ijms-24-10303-f001:**
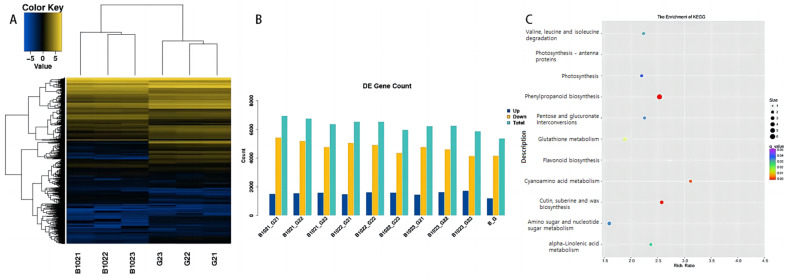
Analysis of differentially expressed genes (DEGs) in young panicles of Guichao 2 and B102 and Kyoto Encyclopedia of Genes and Genomes (KEGG) pathway enrichment map of DEGs. (**A**) Differential gene cluster diagram. (**B**) Counts of up- and down-regulated DEGs. (**C**) KEGG pathway enrichment map. The horizontal axis represents the Rich Ratio. The vertical axis represents the name of the KEGG pathway. The size of the point indicates the number of DEGs in the path. The color of the point represents the enrichment level of each KEGG entry, with colors closer to red indicating higher enrichment levels.

**Figure 2 ijms-24-10303-f002:**
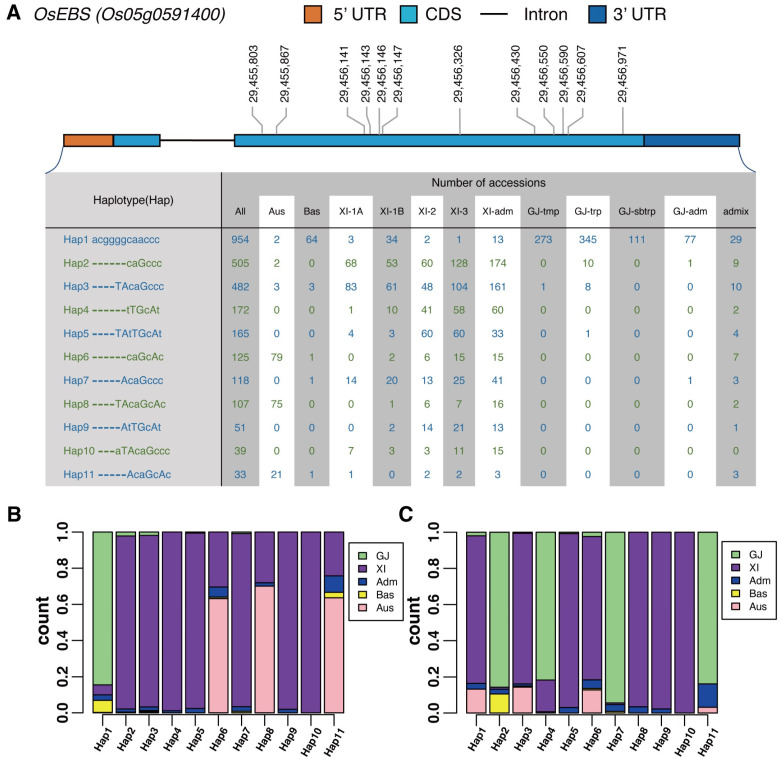
Subpopulation frequency of *OsEBS* haplotypes in 2751 accessions from the 3K-RG database. (**A**) Haplotypes of *OsEBS* (*Os05g0591400*). The haplotype consists of 12 SNPs in the gene-coding sequence (CDS) region. Lowercase letters represent synonymous mutations, whereas uppercase letters indicate non-synonymous mutations. (**B**) The subgroup distribution of CDS haplotypes (gcHaps). (**C**) The subgroup distribution of 1-kb promoter haplotypes (gpHaps).

**Figure 3 ijms-24-10303-f003:**
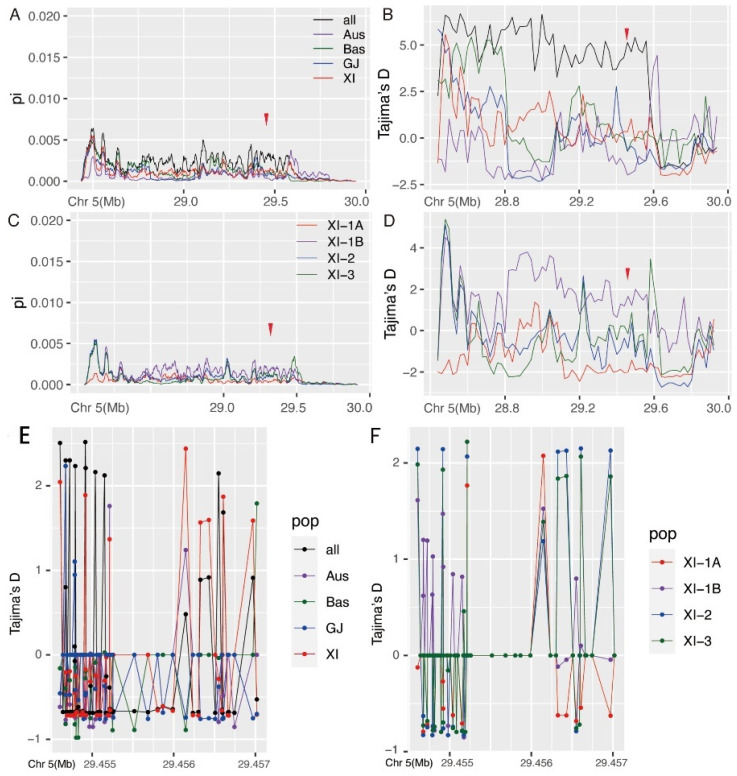
Nucleotide diversity (π) and Tajima’s D analysis of subpopulations. The red arrowheads indicates the position of *OsEBS*. (**A**) Nucleotide diversity (π) for the approximately 2-Mb genomic region flanking *OsEBS* in the four subpopulations: *Aus*, *Bas*, *GJ*, and *XI*. (**B**) Tajima’s D for the approximately 2-Mb genomic region flanking *OsEBS* in the four subpopulations: *Aus*, *Bas*, *GJ*, and *XI*. (**C**) Nucleotide diversity (π) for the approximately 2-Mb genomic region flanking *OsEBS* in the four subpopulations: *XI-1A*, *XI-1B*, *XI-2*, and *XI-3*. (**D**) Tajima’s D for the approximately 2-Mb genomic region flanking *OsEBS in* the four subpopulations: *XI-1A*, *XI-1B*, *XI-2*, and *XI-3*. (**E**) Tajima’s D analysis based on single nucleotide polymorphisms in the region of the upstream 1-kb promoter and the gene regions of *OsEBS* in the four subpopulations: *Aus*, *Bas*, *GJ*, and *XI*. Different colors correspond to different subpopulations as shown in the legend. (**F**) Tajima’s D analysis based on SNP sites in the region of the upstream 1-kb promoter and the gene regions of *OsEBS* in the four subpopulations: *XI-1A*, *XI-1B*, *XI-2*, and *XI-3*. Different colors correspond to different subgroups in the *XI* subpopulation as shown in the legend.

**Figure 4 ijms-24-10303-f004:**
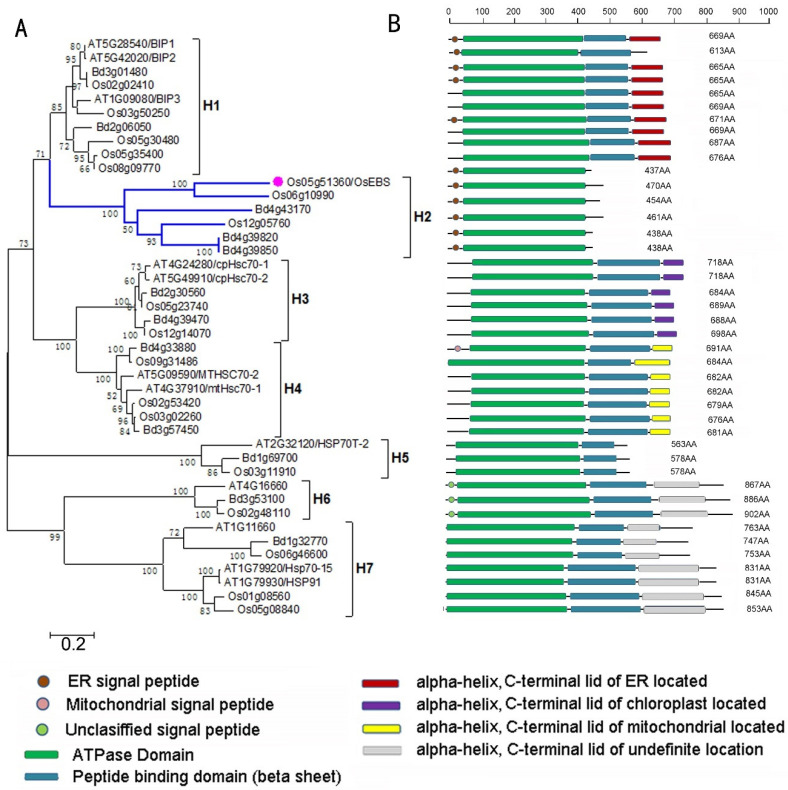
Phylogenetic tree and domain architecture of the Hsp70 family from *Oryza sativa* (Os), *Brachypodium distachyon* (Bd), and *Arabidopsis thaliana* (At). (**A**) Maximum Likelihood (ML) tree constructed using the Jones–Taylor–Thorton (JTT) model with ClustalW2 alignment. Only ML bootstrap values of at least 50% are shown. Scale bar corresponds to 0.2 estimated amino acid substitutions per site. (**B**) Domain architectures of full-length proteins based on InterProScan database searches. Most domain names and diagrams were obtained from SUPERFAMILY and SIGNALP. Names of each domain are shown below the figure. Number of amino acids is shown above the domain architecture. ER, endoplasmic reticulum.

**Table 1 ijms-24-10303-t001:** Statistics and genome comparison of sequencing data for young panicles of Guichao 2 and B102.

Sample	B1021	B1022	B1023	G21	G22	G23
Raw Read Number	41,793,424	46,244,714	48,182,186	40,363,696	45,647,082	39,652,964
Clean Read Number	40,652,450	45,029,452	46,814,038	39,480,362	44,404,198	38,709,036
Clean Read Rate (%)	97.27	97.37	97.16	97.81	97.28	97.62
Clean Base Number	6,097,867,500	6,754,417,800	7,022,105,700	5,922,054,300	6,660,629,700	5,806,355,400
Low-quality Read Number	193,670	222,282	206,996	198,868	456,696	303,188
Low-quality Read Rate (%)	0.46	0.48	0.43	0.49	1.00	0.77
Adapter-Polluted Read Number	946,522	992,144	1,160,242	683,722	785,344	640,076
Adapter-Polluted Read Rate (%)	2.27	2.15	2.41	1.69	1.72	1.61
Raw Q30 Base Rate (%)	94.34	94.12	94.37	93.85	94.05	94.21
Clean Q30 Base Rate (%)	94.54	94.36	94.56	94.08	94.49	94.55

**Table 2 ijms-24-10303-t002:** The significantly enriched biological processes of auxin-related DEGs (q < 0.05).

GO ID	GO Term	DEGs	q Value
GO:0009734	auxin-activated signaling pathway	40	0.0335
GO:0009926	auxin polar transport	16	0.0158
GO:0060918	auxin transport	20	0.0014
GO:0010540	basipetal auxin transport	11	0.0022
GO:0003333	amino acid transmembrane transport	13	0.0014
GO:0055114	oxidation–reduction process	79	0.0004
GO:0009813	flavonoid biosynthetic process	47	0.0000
GO:0015706	nitrate transport	6	0.0465

## Data Availability

The data presented in this study are available upon request from the corresponding authors.

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
