# Peer review of "RNA-Seq Transcriptome Analysis and Evolution of OsEBS, a Gene Involved in Enhanced Spikelet Number per Panicle in Rice"

_ijms, 2023, doi:10.3390/ijms241210303_

Round 1

Reviewer 1 Report

The paper “RNA-Seq Transcriptome Analysis and Evolution of OsEBS, a Gene Involved in Enhanced Spikelet Number Per Panicle in Rice” by Fuan Niu, Mingyu Liu, Shiqing Dong, Xianxin Dong, Ying Wang, Can Cheng, Huangwei Chu, Zejun Hu, Fuying Ma, Peiwen Yan, Dengyong Lan, Jianming Zhang, Jihua Zhou, Bin Sun, Anpeng Zhang, Jian Hu, Xinwei Zhang, Shicong He, Jinhao Cui, Xinyu Yuan, Jinshui Yang, Liming Cao and Xiaojin Luo describes a comprehensive study, in which 173 rice varieties (114 japonica, 55 indica + 4 wild rice) aimed on the increase of spikelet number per panicle, which is crucial factor for the rice yield.

Efficient harvest is very important for major cereals; with high yield crops, enough food can be obtained for global consumption from available fertile land, leaving some of it free to grow other crops to increase food diversity without the need to turn more land into less productive fields.

The paper is well organized into section according to investigated topics: RNA sequencing and sequence alignment, analysis of differentially expressed genes, function enrichment analysis, the search for full length and truncated alleles of OsEBS in indica, japonica and wild subspecies, population genetic analysis and the phylogenetic tree study. The authors proved the hypothesis that the gene responsible for enhanced spikelet number per panicle is rarely used in indica rice and the introgression of the OsEBS full-length allele into high-yielding indica varieties could further improve the yield of indica rice.

The results are given in good, simple scientific English, the amount of tables and charts is appropriate and they are well arranged. Fig. 4B is a bit hard to read in the *.pdf proof and should be copied to the final manuscript from original graphic file. (This is not the fault of authors, text processors tend to make problems with imported graphic and it is safer to place a “hard” picture to reviewers’ version to be sure it will be displayed in the right place).

The study brings a lot of new information, both interesting and beneficial, for the readers of the International Journal of Molecular Sciences, and thus is recommended for acceptation.

However, there is one suggestion to the authors. Some of your observation suggested possible enhanced flavonoid biosynthesis. Rice is a base of food in large part of the world and favourite side dish in the other; it is consumed in large quantities and the amount of health-favourable compounds is rather important. It would be worth a try to monitor also the level of antioxidants to find out if some of the changes in metabolic pathways make some varieties better candidates for flavonoid-rich functional food.

Reviewer 2 Report

Congratulations!

Author Response

Thank you for your recognition of our work.

Reviewer 3 Report

Dear Authors
I have some comments which may improve your manuscript.
Please change the abstract by providing information on the materials used for phylogenetic research.
In the last paragraph of the Introduction (lines 72 - 74), you wrote that SNP increases mechanisms in OsEBS transformants are still unclear, and this aspect was discussed. However, you also wrote that "this study provides a theoretical basis for a high-yield rice breeding strategy and a deeper understanding of neofunctionalization". It is a fact. But it should be concluded in the Discussion somehow.

Please describe more precisely the material for RNA extraction. What BBCH stage was when you collected the plant material? Which part of the panicle was collected? Please use the BBCH scale to describe the work schedule with plant material ( tissue collection, plant transplantation). Some description of weather conditions is needed, at least in Materials and Methods, since environmental conditions influence plant development, and all rice genotypes were cultivated in field conditions. 

Please show the influence of transformation on the phenotype compared to the wild genotype.

I have raised some issues from an agricultural point of view. Please add and discuss the paper published by Sharma et al. in 2018: "Auxin protects spikelet fertility and grain yield under drought and heat stresses in rice" in Environmental and Experimental Botany https://doi.org/10.1016/j.envexpbot.2018.02.013.

In general, auxins are indispensable for good yielding in stress conditions.

Dear Authors, please complete the above information carefully.

I evaluate the manuscript as comprehensive; it presents well the results from statistical calculations and comparisons. I decided to mark "no answer" in the rating part of the evaluation because of the novelty related to the gene studied. However, the gene was discovered in other studies. And the manuscript is a well-done description of the results of statistical analyses.

Anyhow, I recommend the manuscript for publication.

Reviewer

Round 2

Reviewer 3 Report

Dear Authors, thank you for the corrections made.

Reviewer